# Targeting Host PIM Protein Kinases Reduces Mayaro Virus Replication

**DOI:** 10.3390/v14020422

**Published:** 2022-02-18

**Authors:** Madelaine Sugasti-Salazar, Dalkiria Campos, Patricia Valdés-Torres, Paola Elaine Galán-Jurado, José González-Santamaría

**Affiliations:** 1Grupo de Biología Celular y Molecular de Arbovirus, Instituto Conmemorativo Gorgas de Estudios de la Salud, Panama City 0816-02593, Panama; msugasti@gorgas.gob.pa (M.S.-S.); dcampos@gorgas.gob.pa (D.C.); pavt2099@gmail.com (P.V.-T.); paola.elaine.jurado@gmail.com (P.E.G.-J.); 2Programa de Maestría en Microbiología Ambiental, Universidad de Panama, Panama City 3366, Panama

**Keywords:** Mayaro, Una, Zika, host factors, PIM kinases, inhibition, antiviral effect

## Abstract

Mayaro virus (MAYV) manipulates cell machinery to successfully replicate. Thus, identifying host proteins implicated in MAYV replication represents an opportunity to discover potential antiviral targets. PIM kinases are enzymes that regulate essential cell functions and also appear to be critical factors in the replication of certain viruses. In this study we explored the consequences of PIM kinase inhibition in the replication of MAYV and other arboviruses. Cytopathic effects or viral titers in samples from MAYV-, Chikungunya-, Una- or Zika-infected cells treated with PIM kinase inhibitors were evaluated using an inverted microscope or plaque-forming assays. The expression of viral proteins E1 and nsP1 in MAYV-infected cells was assessed using an immunofluorescence confocal microscope or Western blot. Our results revealed that PIM kinase inhibition partially prevented MAYV-induced cell damage and also promoted a decrease in viral titers for MAYV, UNAV and ZIKV. The inhibitory effect of PIM kinase blocking was observed for each of the MAYV strains tested and also occurred as late as 8 h post infection (hpi). Finally, PIM kinase inhibition suppressed the expression of MAYV E1 and nsP1 proteins. Taken together, these findings suggest that PIM kinases could represent an antiviral target for MAYV and other arboviruses.

## 1. Introduction

Mayaro virus (MAYV) is an emerging arbovirus within the Alfavirus genus that belongs to the Togaviridae family [1]. MAYV, along with other alphaviruses, such as Chikungunya (CHIKV) and Una (UNAV), has been grouped within the Semliki Forest complex, based on its antigenic characteristics [2]. These pathogens mainly cause arthritogenic disease with symptoms including fever, myalgia, retro-orbital pain, joint pain, rash, diarrhea, leucopoenia and in some cases, severe polyarthralgia that can last from months to years [3,4].

Several reports indicate that MAYV is actively circulating in different countries of Central and South America, including Panama, Colombia, Peru, Venezuela, Haiti, French Guiana, Argentina, Ecuador, Bolivia and Brazil [5,6]. Although, the transmission of this virus occurs largely through the bites of the sylvatic mosquitoes *Haemagogus janthinomys*, growing evidence suggests that other species of urban vectors, such as *Aedes aegypti* or *Aedes albopictus*, have the potential to spread MAYV, increasing the risk for future epidemics [7,8,9]. Despite this, no protective vaccines or antiviral compounds have been approved to combat MAYV infection. As a result, the search for possible treatments, including antiviral molecules with broad-spectrum activity, remains crucial.

MAYV needs to manipulate a variety of cellular processes to effectively replicate. Therefore, searching for the key host factors implicated in viral replication is an effective strategy for identifying probable antiviral targets [10,11]. Cellular kinases are vital enzymes that regulate an array of important cell activities and also play a crucial role in the replication of diverse families of viruses [12,13,14]. Moreover, it has been suggested that these proteins represent a potential pharmacological target to combat viral infection [14,15]. In line with this, our group recently showed that the inhibition or knockdown of p38 mitogen-activated protein kinase inhibits MAYV replication at least in part through downmodulation of the structural E1 viral protein [16].

The Proviral Integration sites for Moloney murine leukemia virus (PIM) protein family consists of serine/threonine kinases with constitutive activity and includes three members: PIM1, PIM2 and PIM3 [17]. PIM kinases regulate proliferation, survival, metabolism and migration in different cell lines through the phosphorylation of multiple substrates; their upregulation has been observed mainly in hematological malignancies as well as solid tumors, including prostate, pancreatic and colon tumors [17,18,19,20]. Furthermore, aberrant overexpression of PIM kinase members has been associated with poor prognosis in cancer patients [21]. Thus, these enzymes have become an attractive target for oncological drug development programs in the pharmaceutical industry, and some of these compounds have been evaluated in Phase I clinical trials [22,23,24,25]. Various lines of evidence suggest that PIM kinases could be a pro-viral factor for viruses of distinct families and may act through several mechanisms. For instance, in primary B-cells infected with Epstein-Barr virus (EBV), an increase in the *Pim-1* and *Pim-2* gene levels was observed and these oncogenic kinases enhanced the transcriptional activity of EBV EDNA2 protein, probably contributing to B-cell immortalization [26]. In another study, Park and colleagues revealed that PIM1 kinase interacted with non-structural 5A protein from hepatitis C virus (HCV), and this interaction promoted PIM1 protein stability [27]. Moreover, inhibition of all PIM kinase isoforms or knockdown of these enzymes reduced HCV replication in a dose-dependent manner [27]. In addition, inhibition of PIM1 in primary bronchial epithelial cells decreased human rhinovirus 16 (HRV-16) replication, which suggests that PIM1 kinase is relevant to the replication of this pathogen [28]. More recently, using zebrafish as an in vivo infection model, Pereiro and collaborators found that treating with pan-PIM kinase inhibitors SGI-1776, INCB053914 and AZD1208 blocked the replication of spring viraemia of carp virus (SVCV) [29]. Together, these previous observations indicate that PIM kinases may be important factors implicated in viral replication and could serve as potential antiviral targets. However, whether PIM kinases play a key role in MAYV replication has not yet been investigated. In this study, we analyzed the impact of PIM kinase inhibition on the replication of MAYV and other emerging arboviruses.

## 2. Materials and Methods

### 2.1. Cells Lines, Virus Strains and Reagents

Primary human dermal fibroblasts (HDFs) from adults (PCS-201-012), Vero-E6 (CRL-1586), Vero (CCL-81) (all obtained from ATCC, Manassas, VA, USA) and HeLa cells (kindly provided by Dr. Carmen Rivas, CIMUS, Santiago de Compostela, Spain) were grown in Minimal Essential Medium (MEM) or Dulbecco’s Modified Eagle’s Medium (DMEM) supplemented with 10% fetal bovine serum (FBS), 2 mM of _L_-Glutamine and 1% penicillin-streptomycin antibiotic solution (all reagents were obtained from Gibco, Waltham, MA, USA). All cell lines were incubated at 37 °C under a 5% CO_2_ atmosphere. The virus strains used in this study were as follows: Mayaro (MAYV AVR0565, Peru; MAYV TRVL 4675, Trinidad and Tobago; MAYV Guyane, French Guiana) [30]; Una (UNAV, BT-1495-3, Bocas del Toro, Panama) [31] (all previous strains were acquired from the World Reference Center for Emerging Viruses and Arboviruses (WRCEVA) at the University of Texas Medical Branch (UTMB), Galveston, TX, USA); Chikungunya (CHIKV, Panama_256137) [32]; and Zika (ZIKV, 259249). These strains were isolated from patient sera during CHIKV and ZIKV outbreaks in Panama in 2014 and 2015, respectively. All virus strains were propagated, titrated and stored as previously described [11]. The PIM kinase small inhibitory molecules tested were PIM1 inhibitor 2 (PIM1 Inh 2), which blocks the activity of the PIM1 isoform [33], and pan-PIM kinase inhibitor AZD1208, which inhibits all isoforms of the PIM kinase family (PIM1-3) [34]. Both compounds were obtained from Tocris (Minneapolis, MN, USA), dissolved in Dimethyl sulfoxide (DMSO, Sigma-Aldrich, St. Louis, MI, USA) at 10 mM concentration and stored at −20 °C until use. Working solutions of each compound were prepared in MEM or DMEM at the indicated concentrations.

### 2.2. Cell Viability Analysis

Confluent HeLa cells, HDFs or Vero cells grown in 96-well plates were treated with PIM1 Inh 2 or AZD1208 at concentrations of 1, 5 and 10 µM or DMSO (0.1%), which was used as a control. Cells were incubated for 24 or 48 h, and then, 5 mg/mL of 3-(4,5-Dimethyl-2-thiazolyl-2,5-diphenyltetrazolium bromide (MTT, Sigma-Aldrich, St. Louis, MI, USA) solution in PBS was added to the cells and incubated for an additional 4 h. Formazan crystals were dissolved in a solution of 4 mM HCl and 10% Triton X-100 in isopropanol, and absorbance was measured at 570 nm using a microplate spectrophotometer (Bio Tek, Winooski, VT, USA). Results are shown as the percentage of viable cells relative to the DMSO-treated control cells [16].

### 2.3. Evaluation of PIM Kinase Inhibitors’ Antiviral Effects

HeLa cells, HDFs or Vero cells grown in 12-well plates were pre-treated with PIM1 Inh 2 (5 or 10 µM), AZD1208 (5 or 10 µM) or DMSO (0.1%) for 2 h. After this, the compounds were removed and the cells were infected with MAYV, CHIKV, UNAV o ZIKV at an MOI of 1 or 10. After 1 h of virus adsorption, PIM inhibitors or DMSO in fresh medium were added to the cells and incubated for an additional 24 or 48 h. Then, cell supernatants were collected to quantify viral particle production using a plaque assay or the cytopathic effects were evaluated using an inverted microscope and an MCI70-HD camera (Leica, Buffalo Grove, IL, USA). For the virucidal assay, 10^7^ plaque-forming units (PFU) of MAYV were incubated with 10 µM of PIM kinase inhibitors at 37 °C for 2 h. Next, any remaining virus were directly enumerated using plaque-forming assay.

### 2.4. Viral Plaque-Forming Assay

Viral titers in cell supernatants were assessed using a plaque-forming assay as performed previously [11]. Briefly, 10-fold serial dilutions of infected samples were used to infect Vero-E6 (MAYV and UNAV) or Vero (CHIKV and ZIKV) cells grown in 6-well plates and incubated for 1 h at 37 °C. Then, the infected samples were removed and the cells were overlaid with a solution of 1% agar in MEM supplemented with 2% FBS and incubated for 3 days at 37 °C. After that, the agar was removed and the cells were fixed with 4% formaldehyde solution in PBS and stained with 2% crystal violet dissolved in 30% methanol solution. Lastly, the number of plaques was counted, and the viral titers were reported as plaque-forming units per milliliter (PFU/mL).

### 2.5. Time of Addition Experiment

HeLa cells were infected with MAYV at an MOI of 10; after 1 h of virus adsorption (time 0 h post infection (hpi)), we added PIM kinase inhibitors or DMSO at 0, 2, 4 or 8 hpi. Then, we collected cell supernatants at 24 hpi and quantified the viral progeny production using a plaque-forming assay.

### 2.6. Immunofluorescence Assay

HDFs and HeLa cells grown on glass coverslips in 24-well plates were pre-treated with PIM kinase inhibitors or DMSO as previously indicated. Next, the cells were infected with MAYV at an MOI of 1 (HDFs) or 10 (HeLa cells) for 1 h. After that, we removed the inoculum and added PIM kinase inhibitors or DMSO in fresh medium and the cells were incubated at 37 °C for an additional 24 h. Then, the cells were fixed, permeabilized and blocked as previously reported [16]. Next, the cells were stained overnight at 4 °C with E1 or nsP1 primary rabbit antibodies (both antibodies previously validated in our laboratory) [11] followed by Alexa Fluor 568 goat antirabbit secondary antibody (Invitrogen, Carlsbad, CA, USA). Finally, coverslips were mounted on slides with Prolong Diamond Antifade Mountant with DAPI (Invitrogen, Carlsbad, CA, USA), and images were captured with an FV1000 Fluoview confocal microscope (Olympus, Lombard, IL, USA). The images were analyzed with ImageJ software.

### 2.7. Protein Analysis

Protein expression was assessed using Western blot as previously reported [16]. Briefly, protein extracts were obtained from mock- or MAYV-infected HeLa cells or HDFs that were treated or untreated with PIM kinase inhibitors in Laemmli buffer with 10% Dithiothreitol (Bio-Rad, Hercules, CA, USA). Proteins were separated in SDS-PAGE, transferred to nitrocellulose membranes and blocked with a solution of 5% non-fat milk in T-TBS buffer for 30 min. Next, membranes were incubated overnight at 4 °C with the following primary antibodies: rabbit polyclonal anti-E1; rabbit polyclonal anti-nsP1 (both validated in our laboratory) [11]; and mouse monoclonal anti-GAPDH (Cat. # VMA00046, Bio-Rad, Hercules, CA, USA). Then, the membranes were washed three times in T-TBS buffer and incubated with HRP-conjugated goat anti-rabbit secondary antibody (Cat. # 926-80011, LI-COR, Lincoln, NE, USA) at room temperature for 1 h. Lastly, the membranes were incubated with SignalFire^TM^ ECL Reagent (Cell Signaling Technology, Danvers, MA, USA) for 5 min and the chemiluminescent signal was detected with a C-Digit scanner (LI-COR, Lincoln, NE, USA).

### 2.8. Data Analysis

All data were analyzed with the Mann–Whitney test or one-way ANOVA followed by Dunnett’s post hoc test. Data analysis and graphics were completed with GraphPad Prism software version 9.3.0 for Mac. All experiments were performed at least three times with three replicates. For each experiment, the mean and standard deviation are shown.

## 3. Results

### 3.1. PIM1 Inhibitor 2 and AZD1208 Partially Prevent MAYV-Induced Cytopathic Effects and Reduce Viral Progeny Production in HeLa Cells and Human Dermal Fibroblasts (HDFs)

To assess the effect of PIM kinase inhibitors on MAYV replication, we first analyzed the viability of HeLa cells or HDFs treated with increasing concentrations of the PIM1 inhibitor 2 (PIM1 Inh 2) or AZD1208 using the MTT method. PIM1 Inh 2 blocks the activity of the PIM1 isoform, whereas AZD1208 inhibits all isoforms of the PIM kinase family (PIM1-3). We found that, for both compounds, cell viability was higher than 85% after 24 or 48 h of treatment, indicating that the tested doses of the inhibitors and the incubation time were well-tolerated in these cell lines (Figure 1A–D). Previous reports have demonstrated that MAYV, similar to other alphaviruses, induces strong cytopathic effects in different cells lines, including Vero cells and HDFs [10,16]. Thus, we decided to examine whether PIM kinase inhibitors were able to protect HeLa cells or HDFs from MAYV-induced damage. To test this hypothesis, we pretreated both cell lines for 2 h with DMSO or PIM kinase inhibitors and then we removed the compounds and infected the cells with MAYV. As shown in Appendix A, we observed strong cytopathic effects at 48 hpi in MAYV-infected HDFs. In contrast, we noted a partial but substantial protection from these effects (Appendix A, lower panels) in infected-HDFs treated with the PIM kinase inhibitors PIM1 Inh 2 or AZD1208. In the case of HeLa cells, the cytopathic effects at 48 hpi were less pronounced. However, we observed some protection in these cells when treated with the PIM kinase inhibitors (Appendix A, upper panels).

In order to evaluate the impact of PIM kinase inhibitors on MAYV progeny production, we performed a similar experiment in which HeLa cells or HDFs were pretreated with increasing doses of PIM1 Inh 2 or AZD1208 or DMSO for 2 h. We then removed the compounds and infected both cell lines with MAYV. After 1 h of virus adsorption, we added fresh medium with the PIM kinase inhibitors or DMSO; following 24 h of incubation, viral particle yields in cell supernatants were quantified using a plaque assay. This experiment showed that the PIM kinase inhibitors promoted a significant dose-dependent reduction in MAYV titers in both cell lines tested (Figure 1E–H). A nearly 2 log decrease in viral titers was observed at the maximum tested dose of the compounds (10 µM) (Figure 1E–H). Interestingly, PIM1 Inh 2 stimulated a decrease in MAYV titers similar to that observed with AZD1208, suggesting that PIM1 isoform inhibition was sufficient to block MAYV replication (Figure 1E,G). To verify whether the observed effects of PIM kinase inhibitors were due to direct action on MAYV particles, we carried out a virucidal assay in which 10^7^ PFU of MAYV were incubated with PIM kinase inhibitors or DMSO for 2 h at 37 °C. Then, we quantified the virus directly using a plaque assay. Our results indicated that the PIM kinase inhibitors did not have a direct virucidal effect on MAYV (Appendix A). The results of this series of experiments suggest that PIM1 Inh 2 and AZD1208 decrease MAYV replication in a dose-dependent manner.

### 3.2. PIM Kinase Inhibitors Reduce Viral Replication in Multiple MAYV Strains

The preceding experiments were performed using MAYV strain AVR0565 isolated in San Martin, Peru (1995) [30]. To confirm the inhibitory activity of the PIM kinase inhibitors in other MAYV strains, we investigated the effects of these compounds on the replication of the TRVL 4675 strain (Mayaro county, Trinidad and Tobago, 1954) and the Guyane strain (French Guiana, 1996) [30]. Consequently, HeLa cells or HDFs were pre-treated with PIM kinase inhibitors or DMSO and infected as previously described. After 24 h of incubation, viral titers in cell supernatants were quantified using a plaque-forming assay. These analyses revealed that both PIM1 Inh 2 and AZD1208 significantly reduced MAYV viral titers when compared to DMSO-treated cells for both viral strains and both cell lines tested (Figure 2A–H). These results illustrate that PIM kinase inhibitors reduce MAYV replication regardless of the strain being tested.

### 3.3. PIM Kinase Inhibitors Reduce MAYV Replication at Late Stages after Viral Adsorption

Since all of the abovementioned infection experiments were carried out by pretreating cell lines with the PIM kinase inhibitors, we designed an additional procedure to determine if the antiviral effect of these compounds is maintained after virus adsorption. We performed a time addition experiment in which each inhibitor was added to HeLa cells at the indicated times after viral adsorption. The cells were then incubated for 24 h (Figure 3A), and the viral titers in the cell supernatants were measured as previously indicated. As shown in Figure 3, both PIM1 Inh 2 and AZD1208 decreased viral progeny production in HeLa cells when added as late as 8 h after virus adsorption, which indicates that PIM kinase inhibition impairs a key post-entry step in the MAYV life cycle (Figure 3A–C).

### 3.4. PIM Kinase Inhibitors Downmodulate the Expression of MAYV Strutural Protein E1 and Non-Structural Protein 1 (nsP1)

Our previous results suggested that PIM kinase inhibitors affect a crucial post-entry step in the MAYV replication cycle. Thus, we investigated the effect of these compounds on the expression of the viral proteins E1 and nsP1. First, we used an immunofluorescence confocal microscope to evaluate the presence of these viral proteins in HeLa cells or HDFs that were untreated or treated with the PIM kinase inhibitors or a control and incubated for 24 h upon viral infection. These analyses indicated that the PIM kinase inhibitor treatment resulted in an important decrease in both viral proteins in both HeLa cells and HDFs when compared to DMSO-treated cells. (Figure 4A,B). To further characterize the previous findings, we performed an infection kinetics experiment and evaluated the levels of viral proteins at different time intervals using Western blot. In these experiments, we detected a clear decline in the levels of both viral proteins in both cell lines treated with PIM kinase inhibitors (Figure 4C–F). These results indicate that the PIM kinase inhibitors considerably suppress the expression of E1 and nsP1 viral proteins.

### 3.5. PIM Kinase Inhibitors Reduce Viral Replication of Una (UNAV) and Zika (ZIKV) Viruses

Studies evaluating PIM kinase inhibition as an antiviral strategy are limited. However, it has been reported that inhibiting the PIM kinase family affects the replication of HRV-16, HCV and SVCV [27,28,29]. To explore whether the PIM kinase inhibitors are able to affect the replication of other emerging or re-emerging arboviruses, we tested the effect of these compounds on the alphaviruses Chikungunya (CHIKV) and Una (UNAV) and the Flavivirus Zika (ZIKV). We pretreated Vero cells with PIM1 Inh2, AZD1208 or DMSO and then infected the cells with CHIKV, UNAV or ZIKV; after 24 h of incubation, viral titers in cells’ supernatants were evaluated using a plaque assay. As shown in Figure 5, both compounds reduced UNAV and ZIKV replication, whereas with CHIKV, the PIM kinase inhibitors did not display any effect (Figure 5A–F). The observed antiviral effect of the PIM kinase inhibitors on UNAV or ZIKV did not appear to be associated with cell toxicity (Figure 5G). These findings reveal that PIM kinase inhibitors also exhibit blocking activity against UNAV and ZIKV, suggesting that these kinases could represent a potentially broad host antiviral target.

## 4. Discussion

MAYV is a neglected arbovirus with increasing circulation in Central and South American countries [5]. Evidence suggests that urban mosquitoes, such as *Aedes aegypti* or *Aedes albopictus*, have the capacity to transmit this virus [8,9]. As a result, regional health authorities have rising concerns regarding MAYV’s potential to provoke a large-scale epidemic as seen before with CHIKV and ZIKV [35]. Despite this potential, there are no FDA-approved antivirals or vaccines to combat this pathogen.

The identification of host proteins or cellular pathways implicated in MAYV replication represents an opportunity to discover potential antiviral targets [10,11,16]. In this regard, kinases are enzymes that regulate vital cell functions and also appear to play a crucial role in virus replication [12,13]. Furthermore, inhibition of these proteins has been suggested as a therapeutic strategy to fight viral infection [14,15]. Recently, our group found that the inhibition or knockdown of mitogen-activated protein kinase p38 decreased MAYV replication in a dose-dependent manner, suggesting that MAYV is able to exploit kinase activity to favor viral replication [16].

PIM kinases control important cell activities including proliferation, survival, metabolism and migration [17]. Moreover, these kinases are implicated in different pathologic conditions, such as hematological malignancies and solid tumors, and their upregulation has been associated with a poor prognosis in patients [21,23]. Due to these characteristics, PIM kinases have become an attractive target for cancer treatments [22,23]. However, a limited number of studies have analyzed these protein kinases as antiviral targets [27,28,29,36]. For this reason, we decided to evaluate the impact of PIM kinase inhibition on the replication of MAYV and other arboviruses. Our results indicate that treating HeLa cells or HDFs with PIM1 Inh 2, which blocks the activity of the PIM1 isoform, or AZD1208, a pan-PIM kinase inhibitor that inhibits all isoforms (PIM1-3), promoted partial but significant protection from MAYV-induced cytopathic effects. Furthermore, we observed a significant dose-dependent reduction in MAYV progeny production in both cell lines treated with PIM kinase inhibitors when compared to DMSO-treated cells. Interestingly, inhibition of the PIM1 isoform was enough to promote a decrease in MAYV titers, suggesting that this isoform is relevant for MAYV replication. Moreover, the PIM kinase inhibitors’ effects did not seem to be associated with a direct virucidal effect on MAYV particles. To further demonstrate the significance of PIM kinase inhibition on MAYV replication, we analyzed the effect of these compounds on two additional MAYV strains, TRVL 4675 and Guyane. In these experiments, we found a clear dose-dependent decrease in MAYV replication regardless of the virus strain tested. Taken together, these findings suggest that PIM kinases may be relevant factors in MAYV replication. Our results are in agreement with previous reports, which indicate that PIM kinases inhibition can impair virus replication [27,28].

In order to explore whether the anti-MAYV activity of the PIM kinase inhibitors remains after viral adsorption, we performed a time of addition experiment. These analyses revealed that both inhibitors were able to reduce MAYV replication as late as 8 h after viral adsorption in HeLa cells, indicating that PIM kinase inhibition impacts a key post-entry step in the MAYV life cycle. Accordingly, we evaluated the effect of PIM kinase inhibitors on the expression of E1 and nsP1 viral proteins using immunofluorescence and Western blot assays. The results of the confocal microscopy analysis indicate that treatment with the PIM kinase inhibitors provoked an important reduction in the number of cells with MAYV E1 and nsP1 proteins in both cell lines tested. Likewise, an infection kinetics experiment demonstrated that the levels of both viral proteins were reduced in the cells treated with these compounds, suggesting that PIM kinase inhibitors block the expression of MAYV proteins. Similar results were observed in Huh7.5 cells pretreated with the pan-PIM kinase inhibitor SGI-1776, which promoted a dose-dependent reduction of NS5A, NS3 and Core proteins from HCV [27].

Having demonstrated that PIM kinase inhibitors reduce MAYV replication, we decided to assess whether these compounds were able to disturb the replication of other emerging or reemerging arboviruses. Vero cells pretreated with PIM kinase inhibitors or DMSO were infected with CHIKV, UNAV or ZIKV, and after 24 h of incubation, viral titers were determined as previously described. These experiments show that PIM kinase inhibitors also affected UNAV and ZIKV replication, indicating that PIM kinases are potential broad antiviral targets. Recently, Zhou and colleagues demonstrated that ZIKV infection induces a time-dependent increase in PIM1 mRNA and protein and knockdown or inhibition of PIM1 isoform reduces ZIKV replication, providing further evidence that PIM kinases could be relevant factors in viral replication [36]. Moreover, these authors identified PIM1 as a negative regulator of the type I interferon pathway; inhibition of the PIM1 isoform promoted an upregulation of interferon-stimulated genes, including PML, OASL and TRIM5, and blocked ZIKV replication. Although we did not analyze the activation of the interferon pathway after PIM kinase inhibition in our cell infection models, this is a plausible mechanism that could explain the observed inhibitory effect of these compounds on MAYV and UNAV alphaviruses.

While the PIM kinase inhibitors used in the present study have not yet been approved for human use, AZD1208 has shown efficacy in preclinical models of acute myeloid leukemia [37]. Nevertheless, the use of PIM kinase inhibitors for the treatment of viral infections requires additional analyses in animal infection models in order to demonstrate the efficacy and safety of these drugs as antivirals. In conclusion, this study supports, for the first time, that PIM kinases could represent an antiviral target for MAYV and other arboviruses.

## Figures and Tables

**Figure 1 viruses-14-00422-f001:**
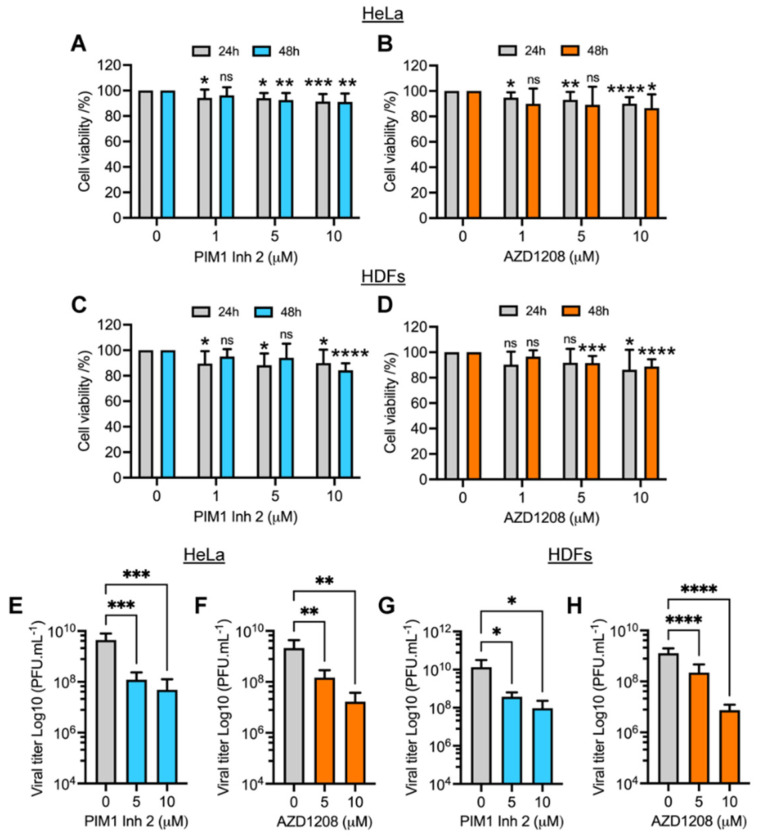
PIM kinase inhibitors PIM1 Inh 2 and AZD1208 reduce MAYV replication in a dose-dependent manner. HeLa cells (**A**,**B**) or HDFs (**C**,**D**) were treated with the indicated concentration of PIM kinase inhibitors and incubated for 24 or 48 h at 37 °C. Next, cell viability was assessed using the MTT method. HeLa cells or HDFs were pretreated with PIM1 Inh 2 (**E**,**G**) or AZD1208 (**F**,**H**) at 5 or 10 µM concentration for 2 h. DMSO was used as a control. Then, the compounds were removed and the cells were infected with MAYV strain AVR0565 at an MOI of 1 (HDFs) or 10 (HeLa cells). After 1 h of virus adsorption, the virus was removed and the cells were treated with PIM kinase inhibitors or DMSO in fresh medium and incubated for an additional 24 h. Finally, viral progeny production in cell supernatants was evaluated using a plaque-forming assay. Viral titers are presented as plaque-forming units per milliliter (PFU/mL). All data were analyzed with the one-way ANOVA test followed by Dunnett’s post hoc test. In the case of cell viability data, this analysis was performed for each drug incubation time. Statistically significant differences are denoted as follows: * *p* < 0.05; ** *p* < 0.01; *** *p* < 0.001; **** *p* < 0.0001; ns: non-significant.

**Figure 2 viruses-14-00422-f002:**
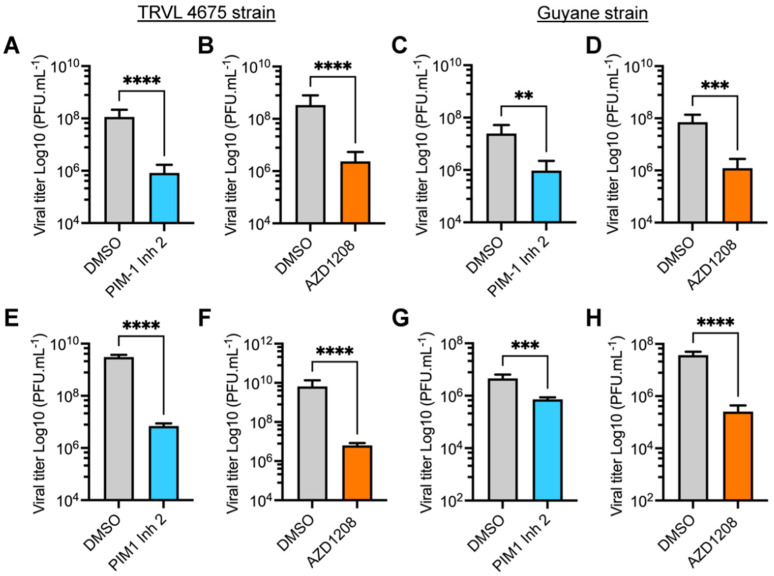
The antiviral effect of PIM kinase inhibitors is independent of the MAYV strain evaluated. (**A**–**D**) HeLa cells or HDFs (**E**–**H**) were pre-treated with PIM kinase inhibitors or DMSO as previously described. Then, the cells were infected with the TRVL 4675 or Guyane MAYV strains and treated with the compounds for 24 h. Next, viral titers in cells supernatants were assessed using a plaque-forming assay. Viral titers are presented as plaque-forming units per milliliter (PFU/mL). Data were analyzed with the Mann–Whitney test. Statistically significant differences are denoted as follows: ** *p* < 0.01; *** *p* < 0.001; **** *p* < 0.0001.

**Figure 3 viruses-14-00422-f003:**
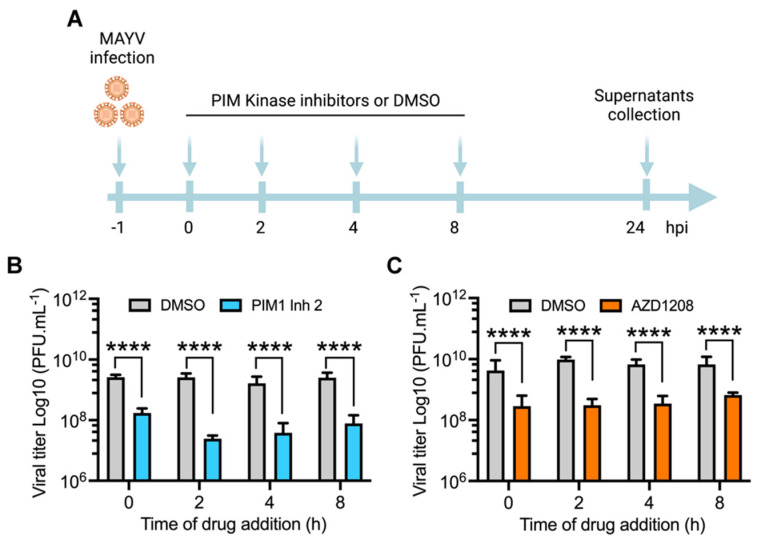
PIM kinase inhibition also affects MAYV replication at a late stage after viral adsorption. (**A**) Diagram showing the design of the time of addition experiment in HeLa cells treated with PIM1 Inh 2 or AZD1208. This panel was created using Biorender.com (accessed on 8 January 2022). (**B**,**C**) HeLa cells were infected with MAYV AVR0565 strain at an MOI of 10. After 1 h of virus adsorption, the cells were treated with PIM kinase inhibitors or DMSO at the indicated times and then incubated at 37 °C for 24 h. Next, viral titers in cells supernatants were quantified using a plaque-forming assay. Viral titers are presented as plaque-forming units per milliliter (PFU/mL). Data were analyzed with the Mann–Whitney test. Statistically significant differences are denoted as follows: **** *p* < 0.0001.

**Figure 4 viruses-14-00422-f004:**
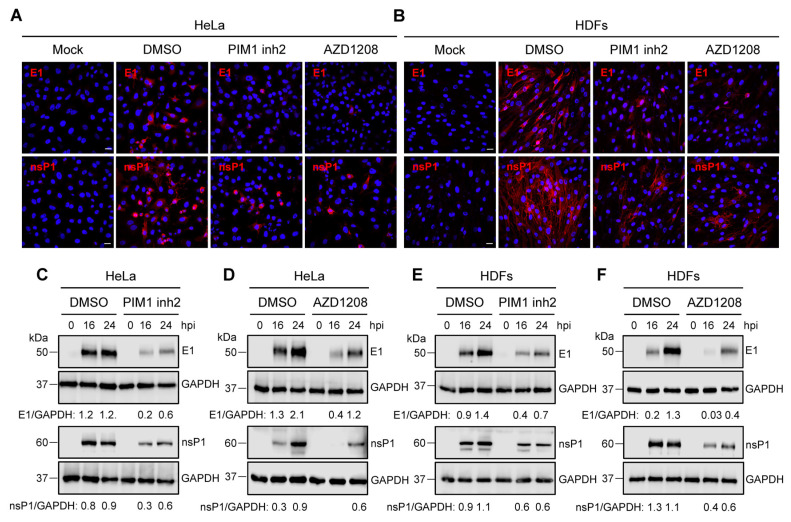
PIM kinase inhibitors reduce the expression of MAYV E1 and nsP1 proteins. HeLa cells (**A**) or HDFs (**B**) were pre-treated with PIM kinase inhibitors or DMSO as previously described. Then, the cells were infected with MAYV AVR0565 strain and after 24 h of infection, the cells positive for E1 and nsP1 proteins were analyzed using an immunofluorescence assay. Scale bar: 20 µm. An infection kinetics experiment was performed with HeLa cells (**C**,**D**) and HDFs (**E**,**F**) that were treated with PIM kinase inhibitors or DMSO to evaluate the levels of MAYV E1 and nsP1 proteins using Western blot. GAPDH protein was used as a loading control. kDa: kilodaltons. Densitometric analysis for E1 and nsP1 proteins was performed using ImageJ software and normalized with GAPDH protein.

**Figure 5 viruses-14-00422-f005:**
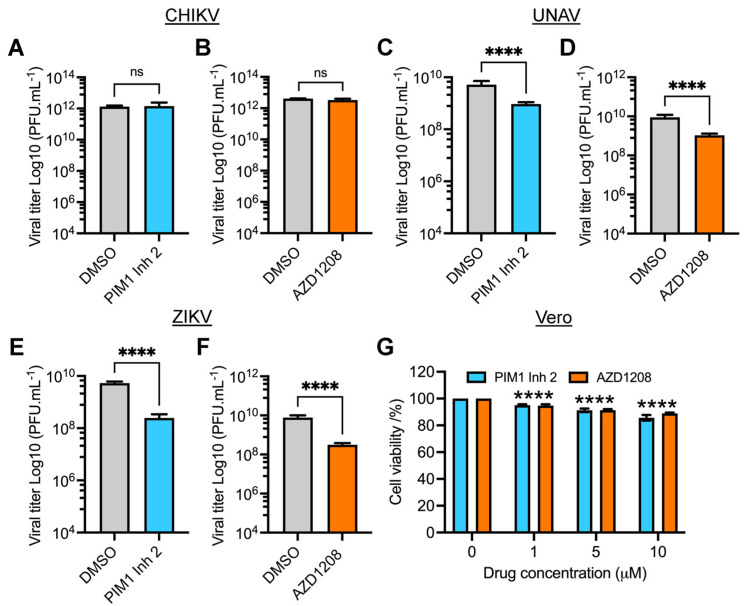
PIM kinase inhibitors block the replication of emerging arboviruses UNAV and ZIKV. Vero cells were pretreated with PIM kinase inhibitors or DMSO for 2 h. Then, the compounds were removed and the cells were infected with CHIKV (**A**,**B**), UNAV (**C**,**D**) or ZIKV (**E**,**F**) at an MOI of 1. After 1 h of virus adsorption, the virus was removed and the cells were treated with PIM kinase inhibitors or DMSO in fresh medium; the cells were then incubated for an additional 24 h. Next, viral titers were quantified as described previously. Viral titers are presented as plaque-forming units per milliliter (PFU/mL). (**G**) Vero cells were treated with the indicated concentrations of PIM1 Inh 2 or AZD1208 for 48 h. DMSO was used as a control. Cell viability was determined using the MTT method as previously performed. Data were analyzed with the Mann–Whitney test or one-way ANOVA test followed by Dunnett’s post hoc test. Statistically significant differences are denoted as follows: **** *p* < 0.0001; ns: non-significant.

## Data Availability

Not applicable.

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
