# Peer review of "Targeting Host PIM Protein Kinases Reduces Mayaro Virus Replication"

_viruses, 2022, doi:10.3390/v14020422_

Round 1

Reviewer 1 Report

Targeting Host PIM Protein Kinases Reduces Mayaro Virus Replication

The authors explored the consequences of PIM kinase inhibition in the replication of MAYV and other arboviruses. Cytopathic effects or viral titers in samples from MAYV-, Una- or Zika-infected cells treated with PIM kinase inhibitors were partially inhibited by up to 3logs. CHIKV was not affected by PIM kinase inhibitor treatment. The inhibitory effect of PIM kinase blocking was observed for different MAYV strains tested and also occurred as late as 8 hpi. Therefore, PIM kinases could represent an antiviral target for MAYV and other arboviruses.

Comments:

The manuscript nicely shows an at least partially inhibition of MAYV, UNA and Zika virus replication by PIM1 inhibitors. Although the mechanism of action is not clear and needs further analyses.

The time of drug addition experiment, suggests that the inhibitors act late. Can the authors exclude that inhibitor is still present in the harvested supernatants and might interfere with plaque formation during titration?

Did the authors analyse interference with interferon and ISG expression? There is a publication about interferon and PIM1 available (DOI: 10.1038/s41392-021-00539-x).

Is RNA synthesis affected by the inhibitors? Is there a difference in inhibition depending on the MOI used for infections?

Is the effect seen after transfection of RNA?

How specific are the PIM kinase inhibitors?

Author Response

Dear Reviewer:

Thank you very much for your comments and suggestions.

We attach a file with our responses.

Best regards,

José González Santamaría, PhD.

Reviewer 2 Report

The work is well designed and relevant.

PIM kinases appear to affect replication, even superficially, i.e. about 1 unit log. It is common to associate more than one antiviral, each one with its effect, even if marginal, adding up its actions in a cocktail.

Treatment with a PIM kinase inhibitor (two forms of inhibitor) did indeed affect virus production and expression of two important virus proteins. In the viral aspect, this is quite evident to me. But for the host cell aspect, I didn't see evidence that the PIM kinases are actually inhibited during the assays. I suggest a simple experiment proving that this inhibition is occurring in the chosen systems and that it is related to low viral yield.

It was discussed that the action of the two inhibitors used is not associated with a direct action on viral particles (virucidal effect). But this action was not excluded by any experiment.

I would like to see a little more review in the introduction ou discussion, for example on the effect of MAP kinase inhibitors (Mitogen Activated Protein Kinases) on viral replication. This is another family of kinases.

In lines 118, 136, 197, 219, 248, 251, 253, 256, 259, 262, 312, 257, 258, the word absorption appears to translate the virus binding to the cell. However, the correct term is adsorption.

Author Response

(The authors gave the same response as above.)
